# A New Limit Theorem for Quantum Walk in Terms of Quantum Bernoulli Noises

**DOI:** 10.3390/e22040486

**Published:** 2020-04-24

**Authors:** Caishi Wang, Suling Ren, Yuling Tang

**Affiliations:** School of Mathematics and Statistics, Northwest Normal University, Lanzhou 730070, China; rslnwnu@163.com (S.R.); tyl0316@163.com (Y.T.)

**Keywords:** quantum walk, quantum Bernoulli noises, limit theorem

## Abstract

In this paper, we consider limit probability distributions of the quantum walk recently introduced by Wang and Ye (C.S. Wang and X.J. Ye, Quantum walk in terms of quantum Bernoulli noises, Quantum Inf. Process. 15 (2016), no. 5, 1897–1908). We first establish several technical theorems, which themselves are also interesting. Then, by using these theorems, we prove that, for a wide range of choices of the initial state, the above-mentioned quantum walk has a limit probability distribution of standard Gauss type, which actually gives a new limit theorem for the walk.

## 1. Introduction

Quantum walks, also known as quantum random walks [1], are quantum analogs of classical random walks, but usually behave quite differently from the classical ones [2]. Due to their wide application in quantum information and quantum computing, quantum walks have received much attention in the past two decades (see e.g., [3,4,5] and references therein). There are two types of quantum walks: discrete-time quantum walks and continuous-time quantum walks. Here we deal with the discrete-time ones only.

Quantum Bernoulli noises refer to the creation and annihilation operators acting on the space H of square-integrable Bernoulli functionals, which satisfy a canonical anti-commutation relation (CAR) in equal time [6]. It has turned out that quantum Bernoulli noises can play a role in describing the irreversible evolution of a quantum system interacting with the environment [7]. So, it is natural to apply quantum Bernoulli noises to the study of quantum walks. In 2016, a model of discrete-time quantum walk on the one-dimensional integer lattice Z was introduced in terms of quantum Bernoulli noises [8], which we call the one-dimensional QBN (Quantum Bernoulli noises) walk below, where QBN is an abbreviation for quantum Bernoulli noises.

Like the usual discrete-time quantum walks, the one-dimensional QBN walk has a unitary representation, hence belongs to the category of unitary quantum walks. However, the one-dimensional QBN walk takes H, the space of square-integrable Bernoulli functionals, as its coin space, hence it has infinitely many internal degrees of freedom since H is infinitely dimensional. More interesting, it has been shown in [8] that, for some very special choices of the initial state, the one-dimensional QBN walk has a limit probability distribution of standard Gauss type.

In this paper, we aim to further examine the limit probability distributions of the one-dimensional QBN walk for a wide range of choices of its initial state. Our motivation comes from the following observations. Let $\Phi_{0}$ be the initial state of the one-dimensional QBN walk. Then $\Phi_{0}\left(0\right)$ is the initial coin state of the walk at position x=0, which completely determines $\Phi_{0}$ since the walk is assumed to start at position x=0. On the other hand, since the coin space H has an orthonormal basis $Z=\{Z_{\sigma }|\sigma \in \Gamma \}$, the initial coin state $\Phi_{0}\left(0\right)$ can be represented as
$$\Phi_{0}\left(0\right)=\sum_{Z_{\sigma }\in Z}\langle Z_{\sigma },\Phi_{0}\left(0\right)\rangle Z_{\sigma },$$
where 〈·,·〉 means the inner product in H and the series on the righthand side converges in the norm. Hence, it is meaningful to make clear the limit probability distribution of the one-dimensional QBN walk with $\Phi_{0}\left(0\right)=\xi$ for a general ξ∈Z.

In the present paper, we first establish several technical theorems, which themselves are also interesting, and then, by using these theorems, we prove that for all ξ∈Z the one-dimensional QBN walk with $\Phi_{0}\left(0\right)=\xi$ still has a limit probability distribution of standard Gauss type (see Theorem 6), which, as our main result, actually establishes a new limit theorem for the one-dimensional QBN walk compared to that of [8].

As is shown in [9], decoherence is one of the important topics in quantum information. Physically, as a dynamical phenomenon, decoherence means a deviation from pure quantum behavior. If the evolution of a quantum system shows up some classical asymptotic behavior, then it contains an amount of decoherence. From a theoretical perspective, it is certainly meaningful to study decoherence in quantum systems including quantum walks [10]. From an application point of view, decoherence can also be useful in quantum walks as was pointed out in [11]. On the other hand, our main result actually implies that, for a wide range of choices of the initial state, the one-dimensional QBN walk has the same limit probability distribution as the classical random walk, which seems to mean that the one-dimensional QBN walk shows up a rather classical asymptotic behavior. Thus, loosely speaking, our main result actually shows that the one-dimensional QBN walk has strong decoherence for a wide range of choices of its initial state.

Recently, a related concept, called quantum Bernoulli factory, has drawn attention (see [12,13] and references therein). The quantum Bernoulli factory essentially stems from quantum walks, hence can be expected to play an important role in the study of quantum walks and their application. It seems that the quantum Bernoulli factory shares some similarities with quantum Bernoulli noises.

The paper is organized as follows. In Section 2, we first briefly recall basic notions and facts about quantum Bernoulli noise, and then describe the one-dimensional QBN walk and its known properties. Our main work then lies in Section 3, where we first establish several technical theorems and then use them to prove the new limit theorem for the one-dimensional QBN walk, among others.

## 2. Preliminaries

In this section, we first briefly recall main notions and facts about quantum Bernoulli noises, and then we describe the one-dimensional QBN walk, which was originally introduced in [8], and its known properties. We refer to [6,7,8] for details.

Throughout this paper, Z always denotes the set of all integers, while N means the set of all nonnegative integers. We denote by Γ the finite power set of N, namely
(1)Γ={σ∣σ⊂N and#σ<∞},
where #σ means the cardinality of σ. Unless otherwise stated, letters like *j*, *k* and *n* stand for nonnegative integers, namely elements of N.

### 2.1. Quantum Bernoulli Noises

Let (Ω,F,P) be a probability measure space and $Z={\left(Z_{n}\right)}_{n\geq 0}$ a sequence of independent random variables on (Ω,F,P), which satisfies that
(1)$F=\sigma (Z_{n};n\geq 0)$, namely F is the σ-field generated by $Z={\left(Z_{n}\right)}_{n\geq 0}$;(2)$P\left\{Z_{n}=\theta_{n}\right\}=p_{n}$, $P\left\{Z_{n}=-1/\theta_{n}\right\}=q_{n}$, n≥0, where $\theta_{n}=\sqrt{q_{n}/p_{n}}$, and ${\left(p_{n}\right)}_{n\geq 0}$ is a given sequence of real numbers with $0<p_{n}<1$ for all n≥0.

As is seen, $Z={\left(Z_{n}\right)}_{n\geq 0}$ is actually a discrete-time Bernoulli process with independent components, which is usually known as a Bernoulli noise and its functionals are known as Bernoulli functionals accordingly. In particular, random variables on (Ω,F,P) are Bernoulli functionals since $F=\sigma (Z_{n};n\geq 0)$.

Let H be the space of square integrable complex-valued Bernoulli functionals, namely
(2)$$H=L^{2}\left(\Omega ,F,P\right).$$

We denote by 〈·,·〉 the usual inner product of the space H, and by ∥·∥ the corresponding norm. We further assume that *Z* has the chaotic representation property [14], which means that the family $Z=\{Z_{\sigma }|\sigma \in \Gamma \}$ forms an orthonormal basis of H, where $Z_{\emptyset }=1$ and
(3)$$Z_{\sigma }=\prod_{j\in \sigma }Z_{j},\;\sigma \in \Gamma ,\;\sigma \neq \emptyset .$$

In the following, we call $Z=\{Z_{\sigma }|\sigma \in \Gamma \}$ the canonical ONB of H. Clearly H is infinitely dimensional since $Z=\{Z_{\sigma }|\sigma \in \Gamma \}$ is countably infinite.

It can be shown that [6], for each k∈N, there exists a bounded operator $\partial_{k}$ on H such that
(4)$$\partial_{k}Z_{\sigma }=1_{\sigma }\left(k\right)Z_{\sigma \setminus k},\;\sigma \in \Gamma ,$$
and its adjoint operator $\partial_{k}^{*}$ satisfies
(5)$$\partial_{k}^{*}Z_{\sigma }=\left[1-1_{\sigma }\left(k\right)\right]Z_{\sigma \cup k}\;\sigma \in \Gamma ,$$
where σ∖k=σ∖{k}, σ∪k=σ∪{k} and $1_{\sigma }\left(k\right)$ is the indicator of σ as a subset of N.

The above operator $\partial_{k}$ and its adjoint $\partial_{k}^{*}$ are usually known as the annihilation and creation operators acting on Bernoulli functionals, respectively.

**Remark** **1.**
*The family ${\left\{\partial_{k}^{*},\partial_{k}\right\}}_{k\geq 0}$ of creation and annihilation operators are called quantum Bernoulli noises.*


One of the important properties that quantum Bernoulli noises admit is that they satisfy the canonical anti-commutation relations (CAR) in equal-time [6]. More precisely, for *k*, l∈N, it holds true that
(6)$$\partial_{k}\partial_{l}=\partial_{l}\partial_{k},\;\partial_{k}^{*}\partial_{l}^{*}=\partial_{l}^{*}\partial_{k}^{*},\;\partial_{k}^{*}\partial_{l}=\partial_{l}\partial_{k}^{*}\;\left(k\neq l\right)$$
and
(7)$$\partial_{k}\partial_{k}=\partial_{k}^{*}\partial_{k}^{*}=0,\;\partial_{k}\partial_{k}^{*}+\partial_{k}^{*}\partial_{k}=I,$$
where *I* is the identity operator on H.

For a nonnegative integer n≥0, one can introduce two operators $L_{n}$ and $R_{n}$ on H as follows:(8)$$L_{n}=\frac{1}{2}\left(\partial_{n}^{*}+\partial_{n}-I\right),\;R_{n}=\frac{1}{2}\left(\partial_{n}^{*}+\partial_{n}+I\right).$$

Clearly, $L_{n}$ and $R_{n}$ are self-adjoint, and moreover they admit the following properties
(9)$$L_{n}^{2}=-L_{n},\;L_{n}R_{n}=R_{n}L_{n}=0,\;R_{n}^{2}=R_{n}.$$

It can be further shown [8] that the operators $L_{n}$, $R_{n}$, n≥0, form a commuting family, namely
(10)$$L_{k}L_{l}=L_{l}L_{k},\;R_{k}L_{l}=L_{l}R_{k},\;R_{k}R_{l}=R_{l}R_{k},\;k,\;l\geq 0.$$

### 2.2. Definition of the One-Dimensional QBN Walk

Let $l^{2}\left(Z,H\right)$ be the space of square summable functions defined on Z and valued in H, namely
(11)$$l^{2}\left(Z,H\right)=\left\{\Phi :Z\to H|\sum_{x=-\infty }^{\infty }{∥\Phi \left(x\right)∥}^{2}<\infty \right\}.$$

Then $l^{2}\left(Z,H\right)$ remains a complex Hilbert space, whose inner product ${\langle \cdot ,\cdot \rangle }_{l^{2}\left(Z,H\right)}$ is given by
(12)$${\langle \Phi ,\Psi \rangle }_{l^{2}\left(Z,H\right)}=\sum_{x=-\infty }^{\infty }\langle \Phi \left(x\right),\Psi \left(x\right)\rangle ,\;\Phi ,\;\Psi \in l^{2}\left(Z,H\right),$$
where 〈·,·〉 denotes the inner product of H as indicated in Section 2.1. By convention, we denote by ${∥\cdot ∥}_{l^{2}\left(Z,H\right)}$ the norm induced by ${\langle \cdot ,\cdot \rangle }_{l^{2}\left(Z,H\right)}$. As usual, a vector $\Phi \in l^{2}\left(Z,H\right)$ is called normalized if ${∥\Phi ∥}_{l^{2}\left(Z,H\right)}=1$.

The following definition describes the one-dimensional QBN walk, which was originally given in [8].

**Definition** **1.**
*The one-dimensional QBN walk is a quantum walk on Z that satisfies the following requirements.*

*The state space of the walk is $l^{2}\left(Z,H\right)$ and its states are represented by normalized vectors in $l^{2}\left(Z,H\right)$.*

*The time evolution of the walk is governed by equation*
(13)$$\Phi_{n+1}\left(x\right)=R_{n}\Phi_{n}\left(x-1\right)+L_{n}\Phi_{n}\left(x+1\right),\;x\in Z,\;n\geq 0,$$
*where $\Phi_{n}\in l^{2}\left(Z,H\right)$ denotes the state of the walk at time n≥0.*



Let ${\left(\Phi_{n}\right)}_{n\geq 0}$ be the state sequence of the one-dimensional QBN walk. Then the function $x\mapsto {∥\Phi_{n}\left(x\right)∥}^{2}$ makes a probability distribution on Z, which is called the probability distribution of the walk at time n≥0. In particular, ${∥\Phi_{n}\left(x\right)∥}^{2}$ is the probability that the quantum walker is found at position x∈Z at time n≥0.

As usual, the one-dimensional QBN walk is assumed to start at position x=0, which implies that its initial state $\Phi_{0}$ is a localized one, namely $\Phi_{0}$ is such that $\Phi_{0}\left(x\right)=0$ for x∈Z with x≠0. Thus $\Phi_{0}\left(0\right)$, which is called the initial coin state at position x=0, plays an important role in investigating the asymptotic behavior of the walk.

It is well known that $l^{2}\left(Z,H\right)≅l^{2}\left(Z\right)⊗H$. This just means that $l^{2}\left(Z\right)$ describes the position of the one-dimensional QBN walk, while H describes its internal degrees of freedom. By convention, H is called the coin space of the one-dimensional QBN walk.

**Remark** **2.**
*The one-dimensional QBN walk has infinitely many internal degrees of freedom, since its coin space H is infinitely dimensional.*


**Lemma** **1.**
*[8] For each n≥0, there exists a unitary operator $U_{n}$ on $l^{2}\left(Z,H\right)$ such that*
(14)$$\left[U_{n}\Phi \right]\left(x\right)=R_{n}\Phi \left(x-1\right)+L_{n}\Phi \left(x+1\right),\;x\in Z,\;\Phi \in l^{2}\left(Z,H\right)$$
*and*
(15)$$\left[U_{n}^{*}\Phi \right]\left(x\right)=R_{n}\Phi \left(x+1\right)+L_{n}\Phi \left(x-1\right),\;x\in Z,\;\Phi \in l^{2}\left(Z,H\right),$$
*where $U_{n}^{*}$ denotes the adjoint of $U_{n}$.*


One can verify that unitary operators $U_{n}$, n≥0, commute mutually, namely $U_{m}U_{n}=U_{n}U_{m}$ for all *m*, n≥0. The next lemma shows that the one-dimensional QBN walk belongs to the category of the so-called unitary quantum walks.

**Lemma** **2.**
*[8] The one-dimensional QBN walk has a unitary representation, more precisely*
(16)$$\Phi_{n+1}=U_{n}\Phi_{n}=\left(\prod_{k=0}^{n}U_{k}\right)\Phi_{0},\;n\geq 0,$$
*where $\Phi_{n}$ is the state of the walk at time n≥0.*


## 3. Main Results

In this section, we show our main work in the present paper. We first establish several technical theorems in Section 3.1, and then in Section 3.2, by using these theorems, we prove our results concerning the limit probability distribution of the one-dimensional QBN walk.

### 3.1. Technical Theorems

Recall that the coin space H has a countably infinite orthonormal basis $Z=\{Z_{\sigma }|\sigma \in \Gamma \}$, which is called the canonical ONB of H.

For k≥0, we write $\Xi_{k}=\partial_{k}^{*}+\partial_{k}$, where $\partial_{k}^{*}$ and $\partial_{k}$ are the creation and annihilation operators on H, respectively (see Section 2.1 for details). It follows from the properties of operators $\partial_{k}^{*}$ and $\partial_{k}$ that $\Xi_{k}$, k≥0 is a commuting sequence of self-adjoint operators on H. Moreover, by the CAR in equal time, one has
$$\Xi_{k}^{2}={\left(\partial_{k}^{*}+\partial_{k}\right)}^{2}=\partial_{k}^{*}\partial_{k}+\partial_{k}\partial_{k}^{*}=I,\;k\geq 0.$$

In the following, we define $\Xi_{\emptyset }=I$ and
(17)$$\Xi_{\sigma }=\prod_{k\in \sigma }\Xi_{k},\;\sigma \in \Gamma ,\;\sigma \neq \emptyset .$$

It can be verified that $\left\{\Xi_{\sigma }|\sigma \in \Gamma \right\}$ forms a commuting family of operators on H.

**Theorem** **1.**
*For any τ, γ∈Γ, it holds that*
(18)$$\Xi_{\tau }Z_{\gamma }=Z_{\tau △\gamma },$$
*where τ△γ=(τ∖γ)∪(γ∖τ), and $Z_{\tau △\gamma }\in Z$ is the basis vector indexed by τ△γ.*


**Proof.** Clearly, $\Xi_{\tau }=\Xi_{\tau \setminus \gamma }\Xi_{\tau \cap \gamma }$. For each k∈τ∩γ, we have
$$\Xi_{k}Z_{\gamma }=\partial_{k}^{*}Z_{\gamma }+\partial_{k}Z_{\gamma }=\left(1-1_{\gamma }\left(k\right)\right)Z_{\gamma \cup k}+1_{\gamma }\left(k\right)Z_{\gamma \setminus k}=Z_{\gamma \setminus k},$$
which implies that $\Xi_{\tau \cap \gamma }Z_{\gamma }=Z_{\gamma \setminus (\tau \cap \gamma )}=Z_{\gamma \setminus \tau }$. On the other hand, for each k∈τ∖γ, we have $\Xi_{k}Z_{\gamma \setminus \tau }=\partial_{k}^{*}Z_{\gamma \setminus \tau }+\partial_{k}Z_{\gamma \setminus \tau }=Z_{(\gamma \setminus \tau )\cup k}$. Thus
$$\Xi_{\tau }Z_{\gamma }=\Xi_{\tau \setminus \gamma }\Xi_{\tau \cap \gamma }Z_{\gamma }=\Xi_{\tau \setminus \gamma }Z_{\gamma \setminus \tau }=Z_{\left(\gamma \setminus \tau )\cup (\tau \setminus \gamma \right)}=Z_{\tau △\gamma },$$
namely (Equation 18) holds. □

Recall that $L_{n}$, $R_{n}$, n≥0 form a commuting sequence of operators on the coin space H, which plays a key role in defining the one-dimensional QBN walk. In what follows, we define $L_{\emptyset }=I$, the identity operator on H, and
(19)$$L_{\sigma }=\prod_{k\in \sigma }L_{k},\;\sigma \in \Gamma ,\;\sigma \neq \emptyset .$$

Similarly we can define $R_{\sigma }$ for any σ∈Γ. It can be verified that $L_{\sigma }$, $R_{\sigma }$, σ∈Γ form a commuting family of self-adjoint operators on H. Additionally, it can be shown that $L_{\sigma }R_{\tau }=0$ whenever σ, τ∈Γ with σ∩τ≠∅.

In what follows, we set $N_{m}=\left\{0,1,\cdots ,m\right\}$ for a nonnegative integer m≥0. Clearly, $N_{m}\in \Gamma$ for all m≥0.

**Theorem** **2.**
*Let m≥0 be a nonnegative integer. Then for each $\sigma \subset N_{m}$, the product $L_{\sigma }R_{N_{m}\setminus \sigma }$ of operators $L_{\sigma }$ and $R_{N_{m}\setminus \sigma }$ has the following representation*
(20)$$L_{\sigma }R_{N_{m}\setminus \sigma }=\frac{1}{2^{m+1}}\sum_{\tau \subset N_{m}}{\left(-1\right)}^{\#(\sigma \setminus \tau )}\;\Xi_{\tau },$$
*where $\sum_{\tau \subset N_{m}}$ means to sum for all subsets τ of $N_{m}$.*


**Proof.** Let $\sigma \subset N_{m}$ and define a function $\varepsilon :N_{m}\to \left\{-1,1\right\}$ as
$$\varepsilon \left(k\right)=\left\{\begin{array}{cc} -1, & k\in \sigma ;\\ 1, & k\notin \sigma . \end{array}\right.$$Then, by a direct computation, we have
$$\begin{array}{cc} L_{\sigma }R_{N_{m}\setminus \sigma }\; & =\prod_{k\in \sigma }\left[\frac{1}{2}\left(\Xi_{k}-I\right)\right]\prod_{k\in N_{m}\setminus \sigma }\left[\frac{1}{2}\left(\Xi_{k}+I\right)\right]\\ \; & =\frac{1}{2^{m+1}}\prod_{k\in \sigma }\left(\Xi_{k}+\varepsilon \left(k\right)I\right)\prod_{k\in N_{m}\setminus \sigma }\left(\Xi_{k}+\varepsilon \left(k\right)I\right)\\ \; & =\frac{1}{2^{m+1}}\prod_{k=0}^{m}\left(\Xi_{k}+\varepsilon \left(k\right)I\right)\\ \; & =\frac{1}{2^{m+1}}\sum_{\tau \subset N_{m}}\left[\prod_{k\in N_{m}\setminus \tau }\varepsilon \left(k\right)\right]\;\Xi_{\tau }. \end{array}$$Note that for each $\tau \subset N_{m}$, it follows easily that
$$\prod_{k\in N_{m}\setminus \tau }\varepsilon \left(k\right)=\prod_{k\in (N_{m}\setminus \tau )\cap \sigma }\left(-1\right)={\left(-1\right)}^{\#(\sigma \setminus \tau )}.$$Thus
$$L_{\sigma }R_{N_{m}\setminus \sigma }=\frac{1}{2^{m+1}}\sum_{\tau \subset N_{m}}\left[\prod_{k\in N_{m}\setminus \tau }\varepsilon \left(k\right)\right]\;\Xi_{\tau }=\frac{1}{2^{m+1}}\sum_{\tau \subset N_{m}}{\left(-1\right)}^{\#(\sigma \setminus \tau )}\;\Xi_{\tau }.$$This completes the proof. □

**Theorem** **3.**
*Let m≥0 be a nonnegative integer. Then, for each $\sigma \subset N_{m}$ and each γ∈Γ, one has*
(21)$${∥L_{\sigma }R_{N_{m}\setminus \sigma }Z_{\gamma }∥}^{2}=\frac{1}{2^{m+1}},$$
*where ∥·∥ denotes the norm in H and $Z_{\gamma }\in Z$ is the basis vector indexed by γ.*


**Proof.** Let $\sigma \subset N_{m}$ and γ∈Γ. Then, by Theorems 1 and 2, we have
(22)$$L_{\sigma }R_{N_{m}\setminus \sigma }Z_{\gamma }=\frac{1}{2^{m+1}}\sum_{\tau \subset N_{m}}{\left(-1\right)}^{\#(\sigma \setminus \tau )}\;Z_{\tau △\gamma }.$$On the other hand, for any $\tau_{1}$, $\tau_{2}\subset N_{m}$ with $\tau_{1}\neq \tau_{2}$, we can find that $\tau_{1}△\gamma \neq \tau_{2}△\gamma$, which implies that $\langle Z_{\tau_{1}△\gamma },Z_{\tau_{2}△\gamma }\rangle =0$. Thus $\left\{Z_{\tau △\gamma }|\tau \subset N_{m}\right\}$ is a finite orthonormal system in H, which together with (Equation 22) yields that
$${∥L_{\sigma }R_{N_{m}\setminus \sigma }Z_{\gamma }∥}^{2}=\frac{1}{2^{2(m+1)}}\sum_{\tau \subset N_{m}}{∥Z_{\tau △\gamma }∥}^{2}=\frac{1}{2^{2(m+1)}}\sum_{\tau \subset N_{m}}1.$$Noting the fact
∑τ⊂Nm1=∑j=0m+1∑#τ=j,τ⊂Nm1=∑j=0m+1m+1j=2m+1,
we finally come to
$${∥L_{\sigma }R_{N_{m}\setminus \sigma }Z_{\gamma }∥}^{2}=\frac{1}{2^{2(m+1)}}2^{m+1}=\frac{1}{2^{m+1}},$$
namely (Equation 21) holds. □

### 3.2. New Limit Theorem

As mentioned above, the initial state $\Phi_{0}$ of the one-dimensional QBN walk is always assumed to be localized, namely $\Phi_{0}\left(x\right)=0$ for all x∈Z with x≠0.

The next lemma originally appeared in [15] without proof. Here, by using Fourier transform for vector-valued functions, we give it a full and rigorous proof.

**Lemma** **3.**
*[15] Let $\left\{\Phi_{n}\right\}$ be the state sequence of the one-dimensional QBN walk with the initial state $\Phi_{0}$. Then, for all n≥1, it holds that*
(23)$$\Phi_{n}\left(x\right)=\left\{\begin{array}{cc} \sum_{\sigma \in \Delta_{j}^{n-1}}L_{\sigma }R_{N_{n-1}\setminus \sigma }\Phi_{0}\left(0\right), & x=n-2j,\;0\leq j\leq n;\\ 0, & otherwise, \end{array}\right.$$
*where $\Delta_{j}^{n-1}=\left\{\sigma \subset N_{n-1}|\#\sigma =j\right\}$.*


**Proof.** Let $\hat{\Phi_{n}}$ be the Fourier transform of $\Phi_{n}$. Then, by the properties of Fourier transforms, we can get
(24)$$\hat{\Phi_{n}}\left(t\right)=\left(e^{it}R_{n-1}+e^{-it}L_{n-1}\right)\hat{\Phi_{n-1}}\left(t\right)\;a.e.\mathrm{in}\;[0,2\pi ],$$
which, together with induction as well as the fact $\hat{\Phi_{0}}\left(t\right)=\Phi_{0}\left(0\right)$, yields
(25)$$\hat{\Phi_{n}}\left(t\right)=\left[\prod_{k=0}^{n-1}\left(e^{it}R_{k}+e^{-it}L_{k}\right)\right]\Phi_{0}\left(0\right)\;a.e.\mathrm{in}\;[0,2\pi ].$$On the other hand, by direct calculation, we find
(26)$$\prod_{k=0}^{n-1}\left(e^{it}R_{k}+e^{-it}L_{k}\right)=\sum_{\sigma \subset N_{n-1}}e^{it(n-2\#\sigma )}L_{\sigma }R_{N_{n-1}\setminus \sigma },$$
which together with (Equation 25) gives
$$\hat{\Phi_{n}}\left(t\right)=\sum_{\sigma \subset N_{n-1}}e^{it(n-2\#\sigma )}L_{\sigma }R_{N_{n-1}\setminus \sigma }\Phi_{0}\left(0\right).$$Thus
$$\Phi_{n}\left(x\right)=\frac{1}{2\pi }\int_{0}^{2\pi }e^{-itx}\hat{\Phi_{n}}\left(t\right)dt=\sum_{\sigma \subset N_{n-1}}\left[\frac{1}{2\pi }\int_{0}^{2\pi }e^{-it(n-2\#\sigma -x)}dt\right]L_{\sigma }R_{N_{n-1}\setminus \sigma }\Phi_{0}\left(0\right),$$
which implies (Equation 23). □

Recall that $L_{\sigma }$, $R_{\sigma }$, σ∈Γ form a commuting family of self-adjoint operators on H. Moreover $L_{\sigma }R_{\tau }=0$ whenever σ, τ∈Γ with σ∩τ≠∅.

**Theorem** **4.**
*Let ${\left\{\Phi_{n}\right\}}_{n\geq 0}$ be the state sequence of the one-dimensional QBN walk. Then, for all n≥1, it holds that*
(27)$${∥\Phi_{n}\left(x\right)∥}^{2}=\left\{\begin{array}{cc} \sum_{\sigma \in \Delta_{j}^{n-1}}{∥L_{\sigma }R_{N_{n-1}\setminus \sigma }\Phi_{0}\left(0\right)∥}^{2}, & x=n-2j,\;0\leq j\leq n;\\ 0, & otherwise, \end{array}\right.$$
*where $\Delta_{j}^{n-1}=\left\{\sigma \subset N_{n-1}|\#\sigma =j\right\}$.*


**Proof.** Let n≥1. We first note that, for σ, $\tau \subset N_{n-1}$ with σ≠τ, one has
$$\sigma \cap \left(N_{n-1}\setminus \tau \right)\neq \emptyset \;\mathrm{or}\;\tau \cap \left(N_{n-1}\setminus \sigma \right)\neq \emptyset ,$$
which together, with the properties of operators $L_{n}$ and $R_{n}$, implies that $L_{\sigma }R_{N_{n-1}\setminus \tau }=0$ or $L_{\tau }R_{N_{n-1}\setminus \sigma }=0$, which then gives
$$\left(L_{\sigma }R_{N_{n-1}\setminus \sigma }\right)\left(L_{\tau }R_{N_{n-1}\setminus \tau }\right)=\left(L_{\sigma }R_{N_{n-1}\setminus \tau }\right)\left(L_{\tau }R_{N_{n-1}\setminus \sigma }\right)=0.$$Now let x=n−2j, 0≤j≤n. Then, by using Lemma 3, we have
$$\begin{array}{cc} {∥\Phi_{n}\left(x\right)∥}^{2}\; & ={∥\sum_{\sigma \in \Delta_{j}^{n-1}}L_{\sigma }R_{N_{n-1}\setminus \sigma }\Phi_{0}\left(0\right)∥}^{2}\\ \; & =\sum_{\sigma \in \Delta_{j}^{n-1}}{∥L_{\sigma }R_{N_{n-1}\setminus \sigma }\Phi_{0}\left(0\right)∥}^{2}+\sum_{\sigma ,\tau \in \Delta_{j}^{n-1}}\langle L_{\sigma }R_{N_{n-1}\setminus \sigma }\Phi_{0}\left(0\right),L_{\tau }R_{N_{n-1}\setminus \tau }\Phi_{0}\left(0\right)\rangle \\ \; & =\sum_{\sigma \in \Delta_{j}^{n-1}}{∥L_{\sigma }R_{N_{n-1}\setminus \sigma }\Phi_{0}\left(0\right)∥}^{2}+\sum_{\sigma ,\tau \in \Delta_{j}^{n-1}}\langle \Phi_{0}\left(0\right),\left(L_{\sigma }R_{N_{n-1}\setminus \sigma }\right)\left(L_{\tau }R_{N_{n-1}\setminus \tau }\right)\Phi_{0}\left(0\right)\rangle \\ \; & =\sum_{\sigma \in \Delta_{j}^{n-1}}{∥L_{\sigma }R_{N_{n-1}\setminus \sigma }\Phi_{0}\left(0\right)∥}^{2}+\sum_{\sigma ,\tau \in \Delta_{j}^{n-1}}0\\ \; & =\sum_{\sigma \in \Delta_{j}^{n-1}}{∥L_{\sigma }R_{N_{n-1}\setminus \sigma }\Phi_{0}\left(0\right)∥}^{2}. \end{array}$$As for the case of x∉{n−2j∣0≤j≤n}, it immediately follows from Lemma 3 that ${∥\Phi_{n}\left(x\right)∥}^{2}=0$. Therefore, formula (Equation 27) holds. □

Recall again that $Z=\{Z_{\sigma }|\sigma \in \Gamma \}$, the canonical ONB of the coin space H, is countably infinite. The following theorem shows that, for all ξ∈Z, the one-dimensional QBN walk with initial coin state $\Phi_{0}\left(0\right)=\xi$ has a binomial distribution.

**Theorem** **5.**
*For all ξ∈Z, the one-dimensional QBN walk with initial coin state $\Phi_{0}\left(0\right)=\xi$ has a probability distribution of the following form*
(28)∥Φn(x)∥2=12nnj,x=n−2j,0≤j≤n;0,otherwise,
*where n≥1 and $\Phi_{n}$ denotes the state of the one-dimensional QBN walk at time n≥1.*


**Remark** **3.**
*It was shown in [8] (see Theorem 4.3 therein) that ${∥\Phi_{n}\left(x\right)∥}^{2}$ has a representation of the following form*
∥Φn(x)∥2=12nnj,x=n−2j,0≤j≤n;0,otherwise
*when the initial state $\Phi_{0}$ satisfies that $\Phi_{0}=\phi$. On the other hand, by the definition of the vector ϕ (see (4.1) of [8]), one can find that $\Phi_{0}=\phi$ if and only if $\Phi_{0}\left(0\right)=Z_{\emptyset }$. Thus, here our Theorem 5 is exactly a generalization of Theorem 4.3 in [8].*


**Proof.** (Proof of Theorem 5). Let ξ∈Z and n≥1. Then there is σ∈Γ such that $\xi =Z_{\sigma }$. By Theorem 4 and the assumption $\Phi_{0}\left(0\right)=\xi$, we have
(29)$${∥\Phi_{n}\left(x\right)∥}^{2}=\left\{\begin{array}{cc} \sum_{\sigma \in \Delta_{j}^{n-1}}{∥L_{\sigma }R_{N_{n-1}\setminus \sigma }Z_{\sigma }∥}^{2}, & x=n-2j,\;0\leq j\leq n;\\ 0, & \mathrm{otherwise}, \end{array}\right.$$On the other hand, for each $\sigma \in \Delta_{j}^{n-1}$, where 0≤j≤n, since $\sigma \subset N_{n-1}$, it follows by using Theorem 3 that
$${∥L_{\sigma }R_{N_{n-1}\setminus \sigma }Z_{\sigma }∥}^{2}=\frac{1}{2^{n}}.$$Thus, for each *j* with 0≤j≤n, by using the equality #Δjn−1=nj, we get
∥Φn(n−2j)∥2=∑σ∈Δjn−1∥LσRNn−1∖σZσ∥2=∑σ∈Δjn−112n=12nnj.This completes the proof. □

By using Theorem 5, we come to the next theorem, which shows that, for all ξ∈Z, the one-dimensional QBN walk with $\Phi_{0}\left(0\right)=\xi$ has a limit probability distribution of standard Gauss type.

**Theorem** **6.**
*For n≥0, let $X_{n}$ be a random variable with a probability distribution given by*
(30)$$P\left\{X_{n}=x\right\}={∥\Phi_{n}\left(x\right)∥}^{2},\;x\in Z,$$
*where $\Phi_{n}$ denotes the state of the one-dimensional QBN walk at time n≥0. Then, for all ξ∈Z, the condition $\Phi_{0}\left(0\right)=\xi$ implies that*
$$\frac{X_{n}}{\sqrt{n}}\Rightarrow N\left(0,1\right),$$
*namely $\frac{X_{n}}{\sqrt{n}}$ converges in law to the standard Gaussian distribution as n→∞.*


**Proof.** The proof is much similar to that of Theorem 4.4 in [8]. We omit it here. □

**Remark** **4.**
*It was shown in [8] (see Theorems 4.4 and 4.5 therein) that $\frac{X_{n}}{\sqrt{n}}\Rightarrow N\left(0,1\right)$ whenever*
$$\Phi_{0}\left(0\right)=Z_{\emptyset }\;or\;\Phi_{0}\left(0\right)=\alpha Z_{\emptyset }+\beta Z_{0},$$
*where α, β are complex numbers with ${\left|\alpha \right|}^{2}+{\left|\beta \right|}^{2}=1$. Compared to this, our Theorem 6 actually offers a new limit theorem for the one-dimensional QBN walk.*


The physical implications of Theorem 6 lie in that the one-dimensional QBN walk shows up a rather classical asymptotic behavior for a wide range of choices of its initial state. Thus, loosely speaking, as a small quantum system, the one-dimensional QBN walk has strong decoherence for a wide range of choices of its initial state.

## 4. Conclusions Remarks

As a dynamic phenomenon, decoherence means a deviation from pure quantum behavior. It is important to study decoherence in quantum walks and quantum walks with decoherence. Several ways have been proposed to introduce decoherence in quantum walks and dynamical properties of the resulting quantum walks have been analyzed accordingly (see, e.g., [9] and references therein for details). Our work in this paper, together with that of [8], actually shows that, as a discrete-time quantum walk on Z, the one-dimensional QBN walk has strong decoherence for a considerable range of choices of its initial state. This might suggest that quantum Bernoulli noises can provide an alternative way to introduce decoherence in quantum walks.

On the other hand, from a purely mathematical point of view, a natural question arises. What limit probability distribution does the one-dimensional QBN walk with $\Phi_{0}\left(0\right)=\xi$ have for a general normalized vector ξ∈H? Such a question is also meaningful from a perspective of the quantum theory. Clearly, our results in this paper, together with those in [8], give a partial answer to the question. A full answer still remains lacking.

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
