# Peer review of "A New Limit Theorem for Quantum Walk in Terms of Quantum Bernoulli Noises"

_entropy, 2020, doi:10.3390/e22040486_

Round 1

Reviewer 1 Report

.

Reviewer 2 Report

In the manuscript, entitled “A New Limit Theorem for Quantum Walk in Terms of Quantum Bernoulli Noises”, Caishi et al. present a new limit for the discrete-time quantum walk induced by one-dimensional quantum Bernoulli noises. The authors establish the whole theory for quantum walk based on quantum Bernoulli noises (QBNs) and propose a new limit theorem for the general initial states. QBN is defined on the creation and annihilation operators applied on the state space of the quantum walk, while one-dimensional QBN walk is defined on the one-dimensional lattice. By using the canonical anti-commutation relations and Fourier transform of the state space of the walk, the authors give a rigorous proof of the new limit theorem they propose. Their proof shows that one-dimensional QBM walk has the same limit probability distributions as the classical counterpart and reveal decoherence in the quantum walk. The new limit theorem of the one-dimensional QBM walk and theoretical analysis are interesting. However, the manuscript is not well written, there are some typos and sentence structure problems needed to be corrected. The article content is also inadequate and not well arranged. In general, I recommend its publication in Entropy after the following comments being well addressed. 1. The authors establish the basic theoretical framework according to their companion work [Quantum Inf. Process. 15 (2016), no. 5, 1897-1908]. However, the authors just copy some arguments in the companion work without changing a word and partly constitute section 2. The content should be paraphrased to avoid plagiarism. 2. The manuscript presents a mathematical work and gives rigorous analysis of the theorems. However, it does not match the style of Entropy so much. The authors should add more explanations about quantum information, rather than giving the mathematical formula only. 3. The differences between the work and [Quantum Inf. Process. 15 (2016), no. 5, 1897-1908] are unclear to me. The innovation of the work seems insufficient. The authors should make more explanations. 4. The authors should draw the figures of the probability distribution for the QBN walk to help the readers understand the limit theorem more intuitively. Besides, it would be interesting to show the relations between the initial states and the probability distributions. The authors should make more explanations and had better draw the figures. 5. The abbreviation QBN lacks definition. 6. There is a related concept, called quantum Bernoulli factory, which essentially stems from a quantum walk [Nat. Commun. 6, 8203 (2015); PRL 117, 010502, (2016)]. The authors can review these works in introduction as applications of the theoretical study.

Round 2

Reviewer 2 Report

In the revised manuscript, the authors have addressed my previous comments. However, this manuscript needs some more improvements before acceptance for publication. Below are some specific comments.

1.      The authors explain too much on the meaning of QBN. In fact, it only needs to be defined on its first occasion in the manuscript.

2.      The manuscript needs further careful editing. The authors should pay attention to the format in formulas.
